# How to GAN away detector effects

**Marco Bellagente[1]\***, **Anja Butter[1]**, **Gregor Kasieczka[2]**,
**Tilman Plehn[1] and Ramon Winterhalder[1]**

**1** Institut für Theoretische Physik, Universität Heidelberg, Germany
**2** Institut für Experimentalphysik, Universität Hamburg, Germany

\* bellagente@thphys.uni-heidelberg.de

## Abstract

LHC analyses directly comparing data and simulated events bear the danger of using first-principle predictions only as a black-box part of event simulation. We show how simulations, for instance, of detector effects can instead be inverted using generative networks. This allows us to reconstruct parton level information from measured events. Our results illustrate how, in general, fully conditional generative networks can statistically invert Monte Carlo simulations. As a technical by-product we show how a maximum mean discrepancy loss can be staggered or cooled.

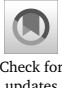

## Content

# 1  Introduction

Our understanding of LHC data from first principles is a unique strength of particle physics. It is based on a simulation chain which starts from a hard process described by perturbative QCD, and then adds the logarithmically enhanced QCD parton shower, fragmentation, hadronization, and finally a fast or complete detector simulation [1]. This simulation chain is publicly available and relies on extremely efficient, fast, and reliable Monte Carlo techniques.

Unfortunately, there is a price for this efficiency: while in principle such a Monte Carlo simulation as a Markov process can be inverted at least statistically, in practice we have to employ approximations. This asymmetry has serious repercussions for LHC analyses, where for instance we do not have access to the likelihood ratio of the hard process. Even worse, it seriously limits our interpretation of LHC results because we cannot easily show results in terms of observables accessible by perturbative QCD. For typical ATLAS or CMS limit reporting this might seem less relevant, but every so often we want to be able to understand such a result more quantitatively.

We propose to use generative networks or GANs [2] to invert Monte Carlo simulations. There are many examples showing that we can GAN such simulations, including phase space integration [3, 4], event generation [5–8], detector simulations [9–15], unfolding [16], and parton showers [17–20]. The question is if and how we can invert them. We start with a naive GAN inversion and see how a mismatch between local structures at parton level and detector level leads to problems. We then introduce the first fully conditional GAN [21] (FCGAN) in particle physics to invert a fast detector simulation [22] for the process

$$pp \to ZW^{\pm} \to (\ell^- \ell^+)\,(jj), \tag{1}$$

as illustrated in Fig. 1. We will see how the fully conditional setup gives us all the required properties of an inverted detector simulation.

We note that our approach is not targeted at combining detector unfolding [23–25] with optimized inference [26–28]. A powerful application for unfolded kinematic distributions to the hard process could be global analyses. For instance in the electroweak and Higgs sector exotics resonance searches turn out to be among the most interesting input and pose a challenge when including them [29]. In contrast, global analyses in the top sector [30–32] successfully rely on unfolded information to different levels of the hard process, for instance the top pair production process [33, 34]. At the same time, alternative methods like simplified template cross sections lose a sizeable amount of information [35]. The same method would also allow us to directly compare first-principles QCD predictions with modern LHC measurements. In addition, our fast inversion might help with advanced statistical techniques like the matrix element method [36–41].

But most importantly, our FCGAN first serves as an example how we can invert Monte Carlo simulations to understand the physics behind modern LHC analyses based on a direct comparison of data and simulations. Here the GAN benefits from the excellent interpolations properties of neural networks. Second, faithfully preserves local structures leading to a large degree of model independence in the unfolding procedure.

# 2  GAN unfolding

A standard method for fast detector simulation is smearing the outgoing particle momenta with a detector response function. This allows us to generate and sample from a probability distribution of smeared final-state momenta for a single parton-level event. For the inversion we need to rely on event samples, as we can see from a simple example: we start from a sharp

$Z$-peak at the parton level and broaden it with detector effects. Now we look at a detector-level event in the tail and invert the detector simulations, for which we need to know in which direction in the invariant mass the preferred value $m_Z$ lies. This implies that unfolding detector effects requires a model hypothesis, which can be thought of as a condition in a probability of the inversion from the detector level. The problem with this point of view is that the parton-level distribution of the invariant mass requires a dynamic reconstruction of the Breit-Wigner peak, which is not easily combined with a Markov process. In any case, from this argument it is clear that unfolding only makes sense at the level of large enough event samples.

For our example we rely on two event samples: we start with events at the parton level, simulated with MADGRAPH5 [42]. For the second sample we first apply PYTHIA8 not including initial state radiation. Technically this means that we only have to deal with a fixed number of partons in the final state and that we can more easily match partons and jets. Further, we apply DELPHES [22] as a fast detector simulation and reconstruct the smeared jet 4-momenta with a jet algorithm included in FASTJET [43]. For lepton 4-momenta we can directly compare the parton-level output with the detector-level output. From Ref. [8] we know how to set up a GAN to either generate detector-level events from parton-level events or vice versa. In our current setup the events are unweighted set of four 4-vectors, two jets and two leptons, but it can be easily adapted to weighted events. The final-state masses are fixed to the parton-level values. Our hadronic final state is defined at the level of jet 4-vectors. This does not mean that in a possible application we take a parton shower at face value. All we do is assume that there is a correspondence between a hard parton and its hadronic final state, and that the parton 4-momentum can be reconstructed with the help of a jet algorithm. The question if for instance an anti-$k_T$ algorithm is an appropriate description of sub-jet physics does not arise as long as the jet algorithm reproduces the hard parton momentum.

Our GAN comprises a generator network $G$ competing against a discriminator network $D$ in a min-max game, as illustrated in Fig. 2. For the implementation we use KERAS (v2.2.4) [44] with a TENSORFLOW (v1.14) backend [45]. As the starting point, $G$ is randomly initialized to produce an output, typically with the same dimensionality as the target space. It induces a probability distribution $P_G(x)$ of a target space element $x$, in our case a parton-level event. To be precise, the generator obtains a batch of detector level event as input and generates a batch of parton level events as output, *i.e.* $G(\{x_d\}) = \{x_G\}$. The discriminator is given batches $\{x_G\}$ and $\{x_p\}$ sampled from $P_G$ and the parton-level target distribution $P_p$. It is trained as a binary classifier, such that $D(x \in \{x_p\}) = 1$ and $D(x) = 0$ otherwise. Following the conventions of Ref. [8] the discriminator loss function is defined as

$$L_D = \langle -\log D(x) \rangle_{x \sim P_p} + \langle -\log(1 - D(x)) \rangle_{x \sim P_G}. \tag{2}$$

We add a regularization and obtain the regularized Jensen-Shannon GAN loss function [46]

$$L_D^{(\text{reg})} = L_D + \lambda_D \left\langle (1 - D(x))^2 |\nabla \phi|^2 \right\rangle_{x \sim P_p} + \lambda_D \left\langle D(x)^2 |\nabla \phi|^2 \right\rangle_{x \sim P_G}, \tag{3}$$

Figure 1: Sample Feynman diagram contributing to $WZ$ production, with intermediate on-shell particles labelled.

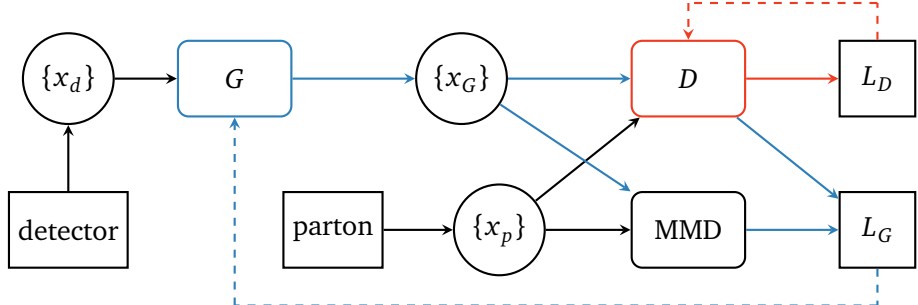

Figure 2: Structure of a naive unfolding GAN. The input $\{x_d\}$ describes a batch of events sampled at detector level and $\{x_{G,p}\}$ denotes events sampled from the generator or parton-level data. The blue (red) arrows indicate which connections are used in the training of the generator (discriminator).

with a properly chosen pre-factor $\lambda_D$ and where we define $\phi(x) = \log \frac{D(x)}{1-D(x)}$. The discriminator training at fixed $P_p$ and $P_G$ alternates with the generator training, which is trained to maximize the second term in Eq.(2) using the truth encoded in $D$. This is efficiently encoded in minimizing

$$L_G = \langle -\log D(x) \rangle_{x \sim P_G}. \tag{4}$$

If the training of the generator and the discriminator with their respective losses Eq.(3) and Eq.(4) is properly balanced, the distribution $P_G$ converges to the parton-level distribution $P_p$, while the optimized discriminator is unable to distinguish between real and generated samples.

If we want to describe phase space features, for instance at the LHC, it is useful to add a maximum mean discrepancy (MMD) [47] contribution to the loss function *. It allows us to compare pre-defined distributions, for instance the one-dimensional invariant mass of an intermediate particle. Given batches of true and generated parton-level events we define the additional contribution to the generator loss as

$$\text{MMD} = \left[ \langle k(x,x') \rangle_{x,x' \sim P_G} + \langle k(y,y') \rangle_{y,y' \sim P_p} - 2\langle k(x,y) \rangle_{x \sim P_G, y \sim P_p} \right]^{1/2}, \tag{5}$$

with another pre-factor $\lambda_G$. Note that we use MMD instead of $\text{MMD}^2$ to enhance the sensitivity of the model [48]. In Ref. [8] we have compared common choices, like Gaussian or Breit-Wigner kernels with a given width $\sigma$,

$$k_{\text{Gauss}}(x,y) = \exp \frac{-(x-y)^2}{2\sigma^2} \qquad \text{or} \qquad k_{\text{BW}}(x,y) = \frac{\sigma^2}{(x-y)^2 + \sigma^2}. \tag{6}$$

As a naive approach to GAN unfolding we use detector-level event samples as generator input. The network input is always a set of four 4-vectors, one for each particle in the final state, with their masses fixed [8]. In the GAN setup we train our network to map detector-level events to parton-level events. Both networks consist of 12 layers with 512 units per layer. With $\lambda_G = 1$, $\lambda_D = 10^{-3}$ and a batch size of 512 events, we run for 1200 epochs and 500 iterations per epoch.

For our $Z_{\ell\ell}W_{jj}$ process we generate 300k events at LO using MADGRAPH5 (v2.6.7) [42] (without any generation cuts) with the standard PYTHIA8 (v8.2) shower [49] and then simulate the detector effects event-by-event with DELPHES (v3.3.3) [22] using the standard ATLAS card. For the reconstruction of the jets we use the anti-$k_t$ jet algorithm [50] with $R = 0.6$ which is

---

*For all details on combining GANs with MMD we refer to the original paper [8].

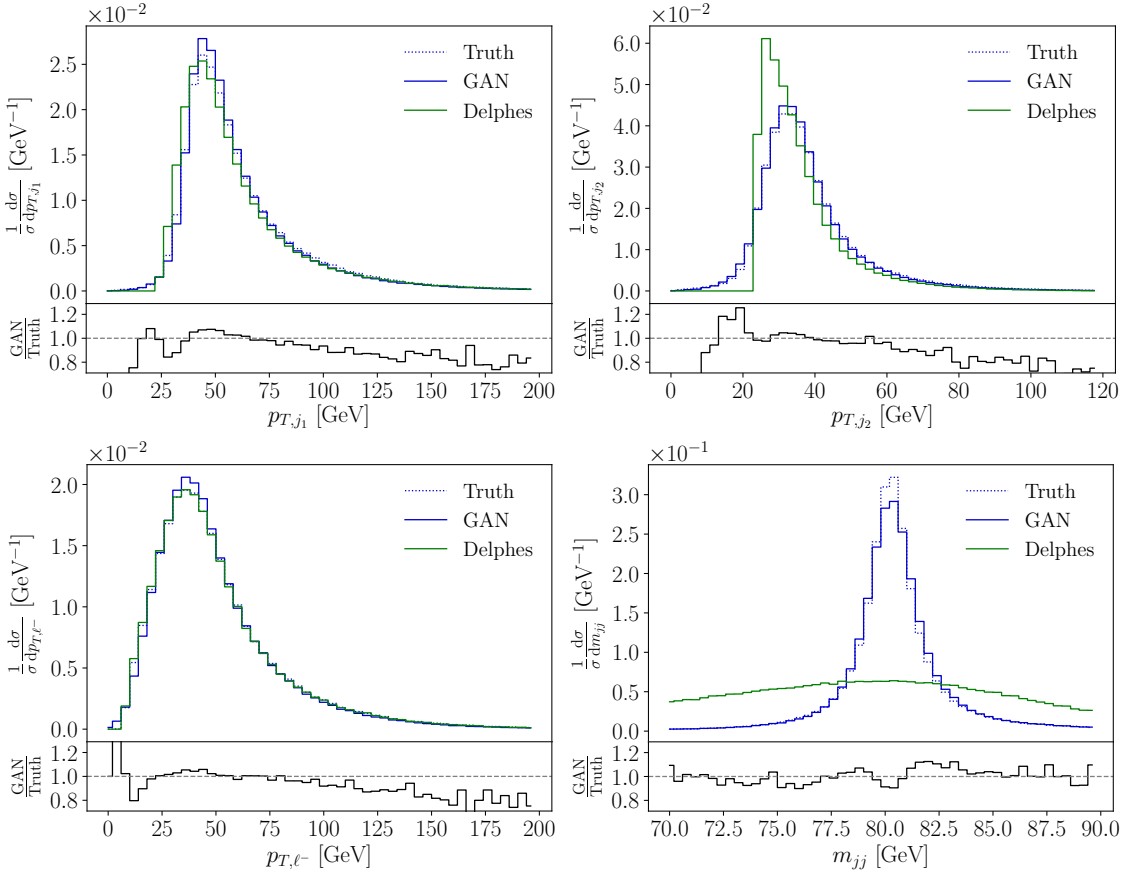

Figure 3: Example distributions for parton level truth, after detector simulation, and GANned back to parton level. The lower panels give the ratio of parton level truth and reconstructed parton level.

performed via the FASTJET (v3.1.3) [43] package included in DELPHES. To keep our toy setup simple we select events with exactly two jets and a pair of same-flavor opposite-sign leptons, specifically electrons. At the detector level both jets are required to fulfill $p_{T,j} > 25$ GeV and $|\eta_j| < 2.5$ GeV. At detector level jets are sorted by $p_T$. We assign each jet to a corresponding parton level object based on their angular distance. The detector and parton level leptons are assigned based on their charge. While the resulting smearing of the lepton momenta will only have a modest effect, the observed widths of the hadronically decaying $W$-boson will be much larger than the parton-level Breit-Wigner distribution. For this reason, we focus on showing hadronic observables to benchmark the performance of our setup.

In Fig. 3 we compare true parton-level events to the output from a GAN trained to unfold the detector effects. We run the unfolding GAN on a set of statistically independent, but simulation-wise identical sets of detector-level events. Both, the relatively flat $p_{T,j_1}$ and the peaked $m_{jj}$ distributions agree well between the true parton-level events and the GAN-inverted sample, indicating that the statistical inversion of the detector effect works well.

A great advantage of this GAN approach is that, strictly speaking, we do not need event-by-event matched samples before and after detector simulation. The entire training is based on batches of typically 512 events, and these batches are independently chosen from the parton-level and detector-level samples. Increasing the batch size within the range allowed by the memory size and hence reducing the impact of event-wise matching will actually improve the GAN training, because it reduces statistical uncertainties [8].

The big challenge arises when we want to unfold an event sample which is not statistically

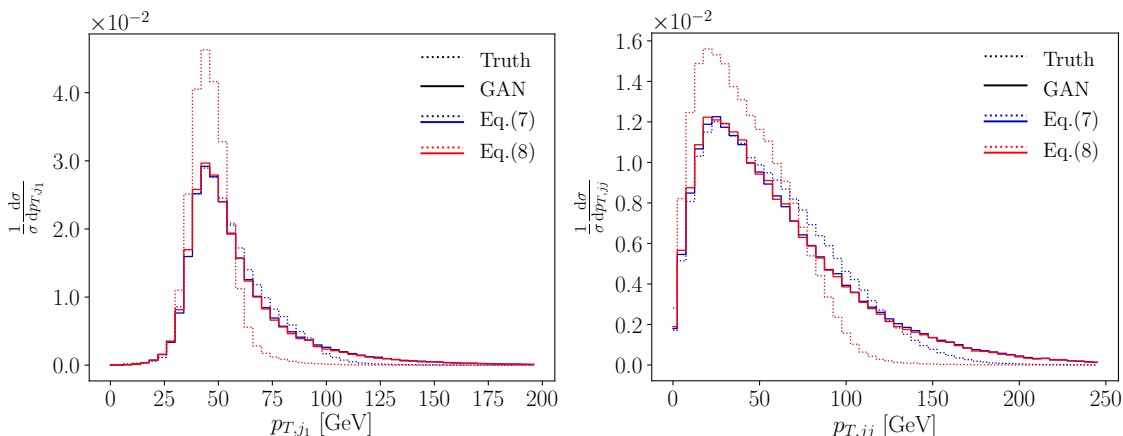

Figure 4: Parton level truth and GANned distributions when we train the GAN on the full data set but only unfold parts of phase space defined in Eq.(7) and Eq.(8).

equivalent to the training data; in other words, the unfolding model is not exactly the same as the test data. As a simple example we train the GAN on data covering the full phase space and then apply and test the GAN on data only covering part of the detector-level phase space. Specifically, we apply the two sets of jet cuts

$$\text{Cut I}: \quad p_{T,j_1} = 30 \dots 100 \text{ GeV} \tag{7}$$

$$\text{Cut II}: \quad p_{T,j_1} = 30 \dots 60 \text{ GeV} \quad \text{and} \quad p_{T,j_2} = 30 \dots 50 \text{ GeV}, \tag{8}$$

which leave us with 88% and 38% of events, respectively. This approach ensures that the training has access to the full information, while the test sample is a significantly reduced sub-set of the full sample.

In Fig. 4 we show a set of kinematic distributions, for which we GAN only part of the phase space. As before, we can compare the original parton-level shapes of the distributions with the results from GAN-inverting the fast detector simulation. We see that especially the GANned $p_{T,j}$ distribution is strongly sculpted by the phase space cuts. This indicates that the naive GAN approach to unfolding does not work once the training and test data sets are not statistically identical. In a realistic unfolding problem we cannot expect the training and test data sets to be arbitrarily similar, so we have to go beyond the naive GAN setup described in Fig. 2. The technical reason for this behavior is that events which are similar or, by some metric, close at the detector level are not guaranteed to be mapped onto events which are close on the parton level. Looking at classification networks this is the motivation to apply variational methods, for instance upgrade autoencoders to variational autoencoders. For a GAN we discuss a standard solution in the next section.

## 3 Fully conditional GAN

The way out of the sculpting problem when looking at different phase space regions is to add a conditional structure to the GAN [21] shown in Fig. 2. The idea behind the conditional setup is not to learn a deterministic link between input and output samples, because we know that without an enforced structure in the weight or function space the generator does not benefit from the structured input. In other words, the network does not properly exploit the fact that the detector-level and parton-level data sets in the training sample are paired. A second, related problem of the naive GAN is that once trained the model is completely deterministic,

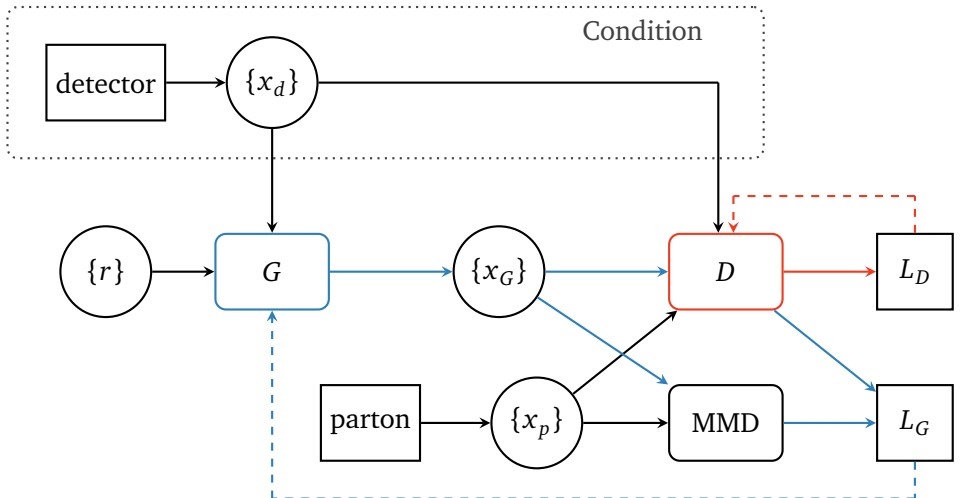

Figure 5: Structure of our fully conditional FCGAN. The input $\{r\}$ describes a batch of random numbers and $\{x_{G,d,p}\}$ denotes events sampled from the generator, detector-level data, or parton-level data. The blue (red) arrows indicate which connections are used in the training of the generator (discriminator).

so each detector-level event will always be mapped to the same parton-level events. This goes against the physical intuition that this entire mapping is statistical in nature.

In Fig. 5 we introduce a fully conditional GAN (FCGAN). It is identical to our naive network the way we train and use the generator and discriminator. However, the input to the generator are actual random numbers $\{r\}$, and the detector-level information $\{x_d\}$ is used as an event-by-event conditional input on the link between a set of random numbers and the parton-level output, *i.e.* $G(\{r\},\{x_d\}) = \{x_G\}$. This way the FCGAN can generate parton-level events from random noise but still using the detector-level information as input. To also condition the discriminator we modify its loss to

$$L_D \to L_D^{(\text{FC})} = \langle -\log D(x,y) \rangle_{x \sim P_p, y \sim P_d} + \langle -\log(1 - D(x,y)) \rangle_{x \sim P_G, y \sim P_d}, \qquad (9)$$

and the regularized loss function changes accordingly,

$$\begin{aligned} L_D^{(\text{reg})} \to L_D^{(\text{reg, FC})} = L_D^{(\text{FC})} &+ \lambda_D \left\langle (1 - D(x,y))^2 |\nabla\phi|^2 \right\rangle_{x \sim P_p, y \sim P_d} \\ &+ \lambda_D \left\langle D(x,y)^2 |\nabla\phi|^2 \right\rangle_{x \sim P_G, y \sim P_d}, \end{aligned} \qquad (10)$$

again using the conventions of Ref. [8]. The generator loss function now takes the form

$$L_G \to L_G^{(\text{FC})} = \langle -\log D(x,y) \rangle_{x \sim P_G, y \sim P_d}. \qquad (11)$$

Table 1: FCGAN setup.

| Parameter | Value | Parameter | Value |
|---|---|---|---|
| Layers | 12 | Batch size | 512 |
| Units per layer | 512 | Epochs | 1200 |
| Trainable weights G | 3M | Iterations per epoch | 500 |
| Trainable weights D | 3M | Number of training events | $3 \times 10^5$ |
| $\lambda_G$ | 1 | | |
| $\lambda_D$ | $10^{-3}$ | | |

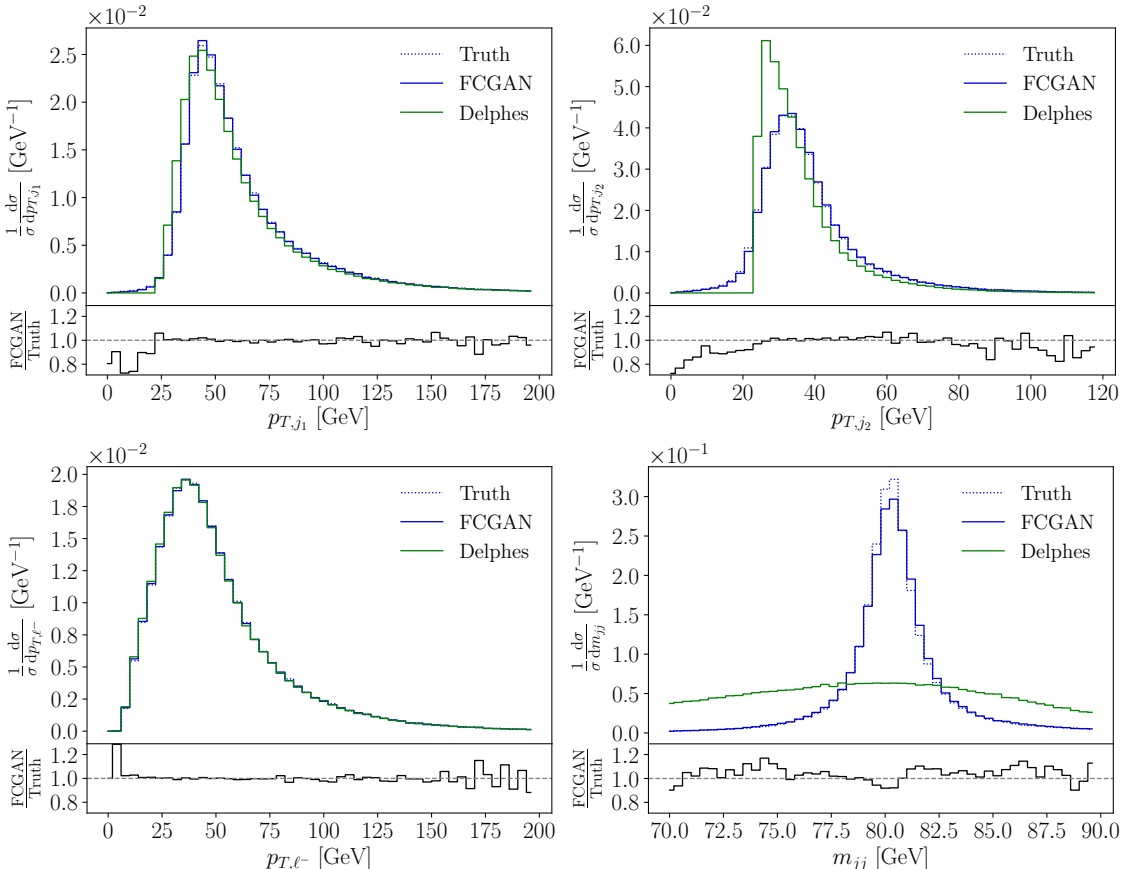

Figure 6: Example distributions for parton level truth, after detector simulation, and FCGANned back to parton level. The lower panels give the ratio of parton level truth and reconstructed parton level. The lower panels give the deviation between parton level truth and reconstructed parton level. To be compared with the naive GAN results in Fig. 3.

Note, that we do not build a conditional version of the MMD loss. The hyper-parameters of our FCGAN are summarized in Tab. 1. Changing from a naive GAN to a fully conditional GAN we have to pay a price in the structure of the training sample. While the naive GAN only required event batches to be matched between parton level and detector level, the training of the FCGAN actually requires event-by-event matching.

In Fig. 6 we compare the truth and the FCGANned events, trained on and applied to events covering the full phase space. Compared to the naive GAN, inverting the detector effects now works even better. The systematic under-estimate of the GAN rate in tails no longer occurs for the FCGAN. The reconstructed invariant $W$-mass forces the network to dynamically generate a very narrow physical width from a comparably broad Gaussian peak. Using our usual MMD loss developed in Ref. [8] we reproduce the peak position, width, and peak shape to about 90%. We emphasize that the MMD loss requires us to specify the relevant one-dimensional distribution, in this case $m_{jj}$, but it then extracts the on-shell mass or width dynamically. The multi-kernel approach we use in this case is explained in the Appendix.

As for our naive ansatz we now test what happens to the network when the training data and the test data do not cover the same phase space region. We train on the full set of events, to ensure that the full phase space information is accessible to the network, but we then only apply the network to the 88% and 38% of events passing the jet cuts I and II defined in Eq.(7) and Eq.(8). We show the results in Fig. 7. As observed before, especially the jet cuts with only

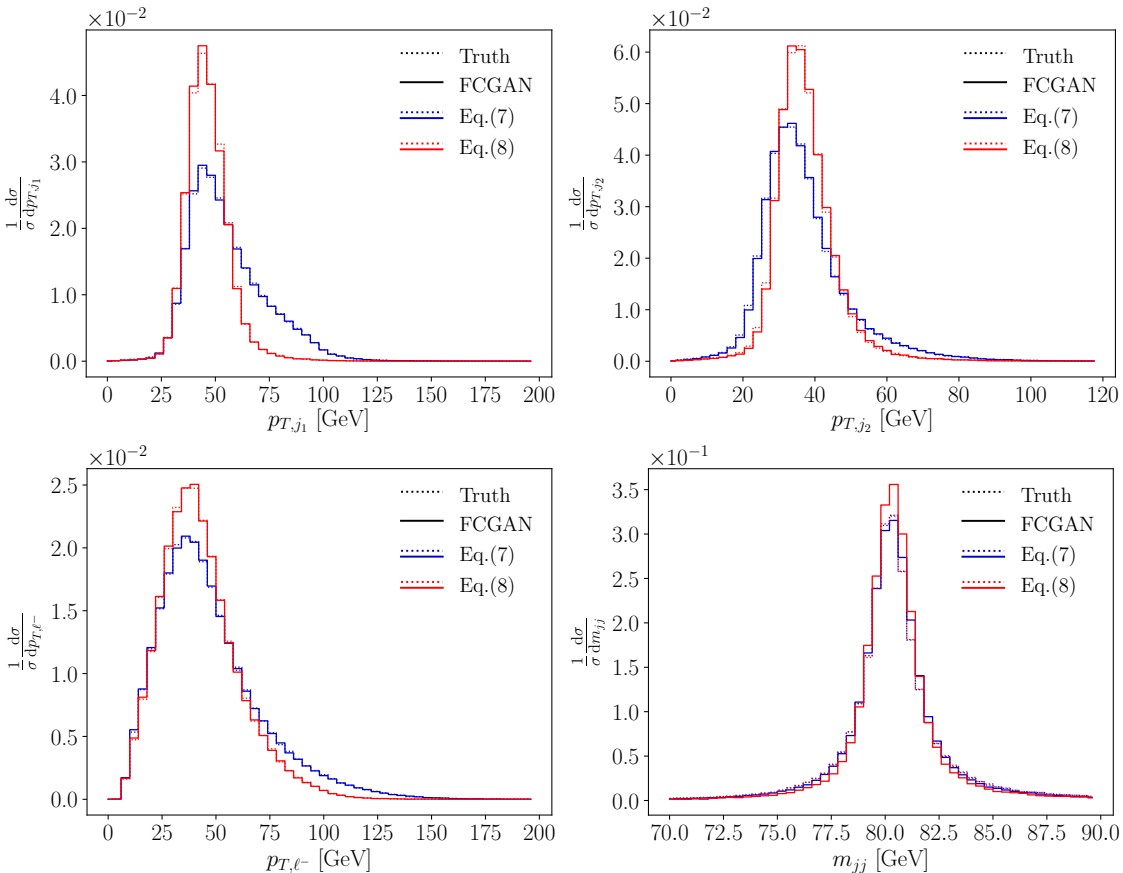

Figure 7: Parton level truth and FCGANned distributions when we train the GAN on the full data set but only unfold parts of phase space defined in Eq.(7) and Eq.(8). To be compared with the naive GAN results in Fig.4.

40% survival probability shape our four example distributions. However, we see for example in the $p_{T,jj}$ distribution that the inverted detector-level sample reconstructs the patterns of the true parton-level events perfectly. This comparison indicates that the FCGAN approach deals with differences in the training and test samples very well.

Because physicists and 4-year olds follow a deep urge to break things we move on to harsher cuts on the inclusive event sample. We start with

$$\text{Cut III}: \quad p_{T,j_1} = 30 \ldots 50 \text{ GeV} \quad p_{T,j_2} = 30 \ldots 40 \text{ GeV} \quad p_{T,\ell^-} = 20 \ldots 50 \text{ GeV}, \quad (12)$$

which 14% of all events pass. In Fig. 8 we see that also for this much reduced fraction of test events corresponding to the training sample the FCGAN inversion reproduces the true distributions extremely well, to a level where it appears not really relevant what fraction of the training and test data correspond to each other.

Finally, we apply a cut which not only removes a large fraction of events, but also cuts into the leading peak feature of the $p_{T,j_1}$ distribution and removes one of the side bands needed for an interpolation,

$$\text{Cut IV}: \quad p_{T,j_1} > 60 \text{ GeV}. \quad (13)$$

For this choice 39% of all events pass, but we remove all events at low transverse momentum, as can be seen from Fig. 6. This kind of cut could therefore be expected to break the unfolding. Indeed, the red lines in Fig. 8 indicate that we have broken the $m_{jj}$ reconstruction through

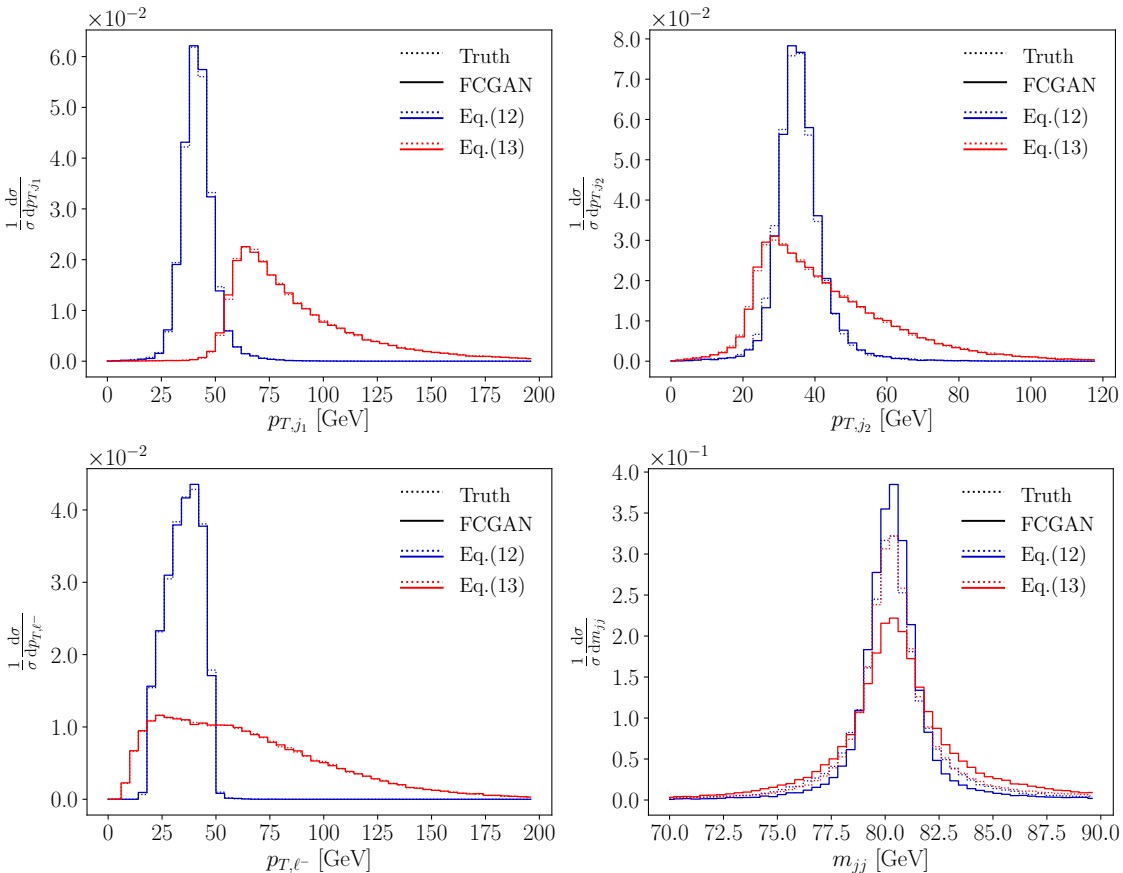

Figure 8: Parton level truth and FCGANned distributions when we train the GAN on the full data set but only unfold parts of phase space defined in Eqs.(12) and (13).

the FCGAN. However, all other (shown) distributions still agree with the parton-level truth extremely well. The problem with the invariant mass distribution is that our implementation of the MMD loss is not actually conditional. This can be changed in principle, but standard implementations are somewhat inefficient and the benefit is not obvious at this stage. At this stage it means that, when pushed towards it limits, the network will first fail to reproduce the correct peak width in the $m_{jj}$ distribution, while all other kinematic variables remain stable.

Finally, just like in Ref. [8] we show 2-dimensional correlations in Fig. 9. We stick to applying the network to the full phase space and show the parton level truth and the FCGAN-inverted events in the two upper panels. Again, we see that the FCGAN reproduces all features of the parton level truth with high precision. The bin-wise relative deviation between the two 2-dimensional distributions only becomes large for small values of $E_{j_1}$, where the number of training events is extremely small.

# 4 New physics injection

As discussed before, unfolding to a hard process is necessarily model-dependent. Until now, we have always assumed the Standard Model to correctly describe the parton-level and detector-level events. An obvious question is what happens if we train our FCGAN on Standard Model data, but apply it to a different hypothesis. This challenge becomes especially interesting if this alternative hypothesis differs from the Standard Model in a local phase space effect.

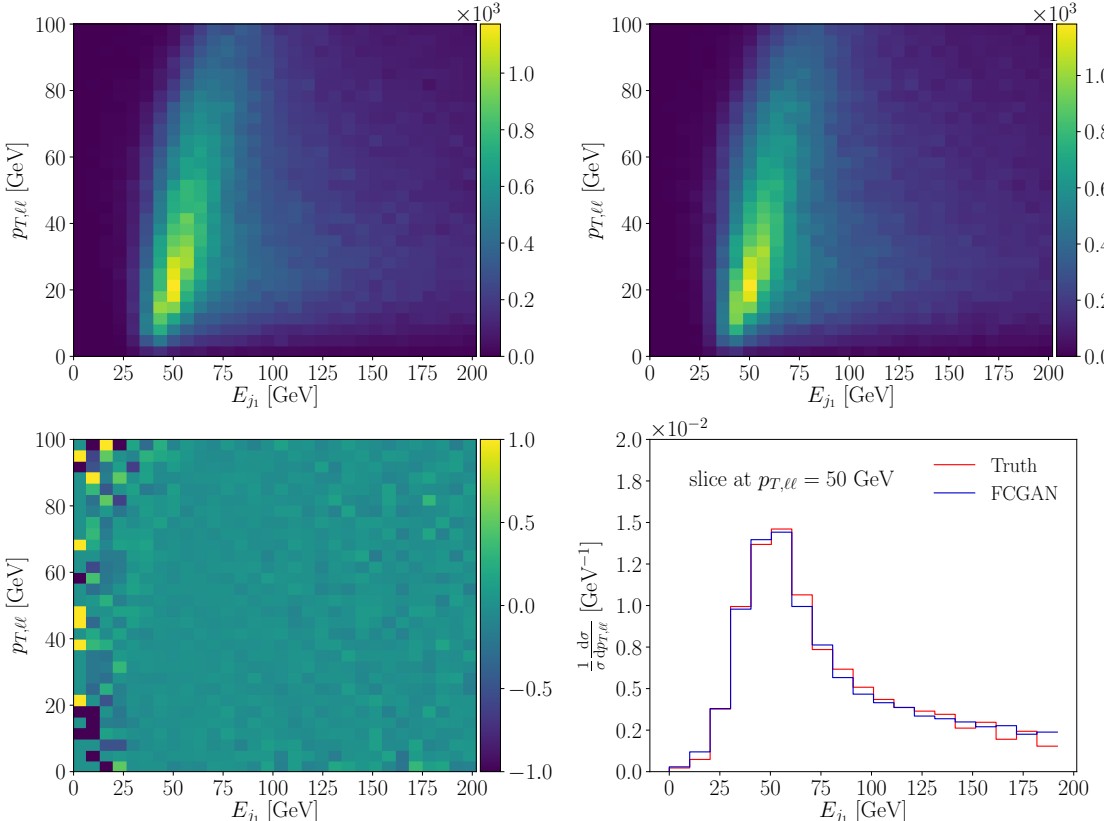

Figure 9: Two-dimensional parton level truth (upper left) and FCGANned (upper right) distributions when we train the GAN on the full data set and unfold over the full phase space. The lower panels show the relative deviation between truth and FCGANned and the one-dimensional $E_{j_1}$ distribution along fixed $p_{T,\ell\ell}$.

It then allows us to test if the generator networks maps the parton-level and detector-level phase spaces in a structured manner. Such features of neural networks are at the heart of all variational constructions, for instance variational autoencoders which are structurally close to GANs. Observing them for GAN unfolding could turn into a significant advantage over alternative unfolding methods.

To this end we add a fraction of resonant $W'$ events from a triplet extension of the Standard Model [51], representing the hard process

$$pp \to W'^* \to ZW^{\pm} \to (\ell^-\ell^+)\,(jj), \tag{14}$$

to the test data. We simulate these events with MADGRAPH5 using the model implementation of Ref. [52] and denote the new massive charged vector boson with a mass of 1.3 TeV and a width of 15 GeV as $W'$. For the test sample we combine the usual Standard Model sample with the $W'$-sample in proportions $90\% - 10\%$. The other new particles do not appear in our process to leading order. Because we want to test how well the GAN maps local phase space structures onto each other, we deliberately choose a small width $\Gamma_{W'}/M_{W'} \sim 1\%$, not exactly typical for such strongly interacting triplet extensions.

The results for this test are shown in Fig. 10. First, we look at transverse momentum distribution of final-state particles, which are hardly affected by the new heavy resonance. Both, the leading jet and the lepton distributions are essentially identical for both truth levels and the FCGAN output. The same is true for the invariant mass of the hadronically decaying $W$-boson, which nevertheless provides a useful test of the stability of our training and testing.

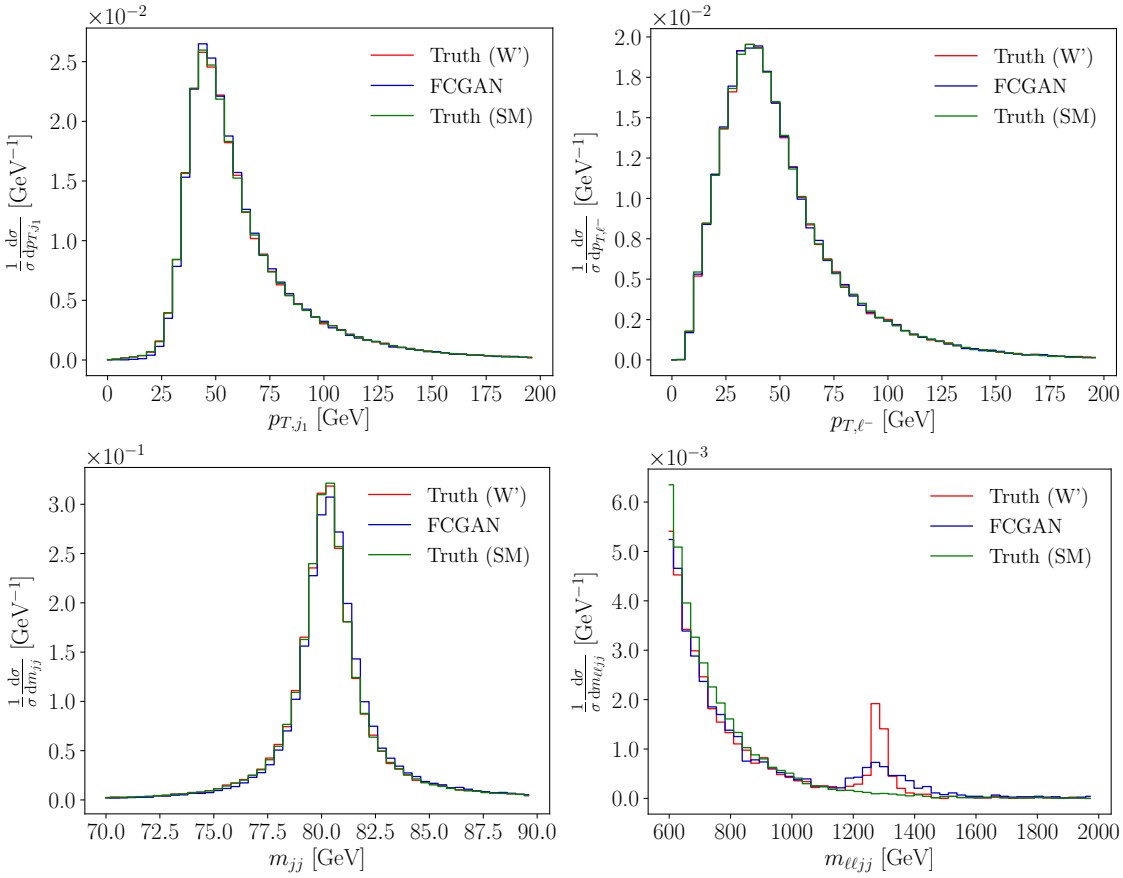

Figure 10: Parton level truth and FCGANned distributions when we train the network on the Standard Model only and unfold events with an injection of 10% $W'$ events. The mass of the additional $s$-channel resonance is 1.3 TeV.

Finally, we show the reconstructed $W'$-mass in the lower-right pane. Here we see the different (normalized) truth-level distributions for the Standard Model and the $W'$-injected sample. The FCGAN, trained on the Standard Model, keeps track of local phase space structures and reproduces the $W'$ peak faithfully. It also learn the $W'$-mass as the central peak position very well. The only issue is the $W'$-width, which the network over-estimates. However, we know already that dynamically generated width distributions are a challenge to GANs and require for instance an MMD loss. Nevertheless, Fig. 10 clearly shows that GAN unfolding shows a high degree of model independence, making use of local structures in the mapping between the two phase spaces. We emphasize that the additional mass peak in the FCGANned events is not a one-dimensional feature, but a localized structure in the full phase space. This local structure is a feature of neural networks which comes in addition to the known strengths in interpolation.

## 5  Outlook

We have shown that it is possible to invert a simple Monte Carlo simulation, like a fast detector simulation, with a fully conditional GAN. Our example process is $WZ \to (jj)(\ell\ell)$ at the LHC and we GAN away the effect of standard DELPHES. A naive GAN approach works extremely well when the training sample and the test sample are very similar. In that case the GAN benefits from the fact that we do not actually need an event-by-event matching of the parton-

level and detector-level samples.

If the training and test samples become significantly different we need a fully conditional GAN to invert the detector effects. It maps random noise parton-level events with conditional, event-by-event detector-level input and learns to generate parton-level events from detector-level events. First, we noticed that the FCGAN with its structured mapping provides much more stable predictions in tails of distributions, where the training sample is statistics limited. Then, we have shown that a network trained on the full phase space can be applied to much smaller parts of phase space, even including cuts in the main kinematic features. The FCGAN successfully maintains a notion of events close to each other at detector level and at parton level and maps them onto each other. This approach only breaks eventually because the MMD loss needed to map narrow Breit-Wigner propagators is not (yet) conditional in our specific setup.

Finally, we have seen that the network reproduces an injected new physics signal as a local structure in phase space. This large degree of model independence reflects another beneficial feature of neural networks, namely the structured mapping of the linked phase spaces.

**FCGAN vs OmniFold**

While we were finalizing our paper, the OMNIFOLD approach appeared [28]. It aims at the same problem as our FCGAN, but as illustrated in Fig. 11 it is completely complementary. Our FCGAN uses the simulation based on DELPHES to train a generative network, which we can apply to LHC events to generate events describing the hard process. The OMNIFOLD approach also starts from matched simulated events, but instead of inverting the detector simulation it uses machine learning to iteratively translate each side of this link to the measured events. This way both approaches should be able to extract hard process information from LHC events, assuming that we understand the relation between perturbative QCD predictions and Monte Carlo events.

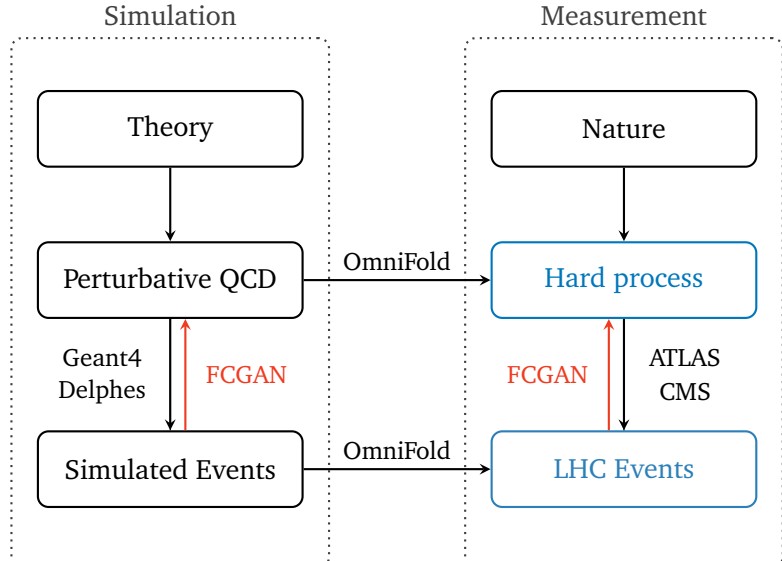

Figure 11: Illustration of the complementary 1.FCGAN and OMNIFOLD [28] approaches.

# Acknowledgments

We would like to thank Lynton Ardizzone for his extremely helpful advice, Ben Nachman for great discussions, and Hans-Christian Schultz-Coulon for the experimental encouragement. Concerning the updated version, we would like to thank Ben Nachman for suggesting a closure test like the one we are showing in Sec. 4 and Johann Brehmer for providing the MADGRAPH5 implementation. RW and MB acknowledge support by the IMPRS-PTFS. The research of AB and MB is supported by the Deutsche Forschungsgemeinschaft (DFG, German Research Foundation) under grant 396021762 – TRR 257 *Particle Physics Phenomenology after the Higgs Discovery*. GK acknowledges support by the Deutsche Forschungsgemeinschaft under grant 390833306 – EXC 2121 *Quantum Universe*.

# A  Performance

While it is clear from the main text that the FCGAN inversion of the fast detector simulation works extremely well, we can still show some additional standard measures to illustrate this. For instance, in Fig. 12 we show the event-wise normalized deviation between the parton-level

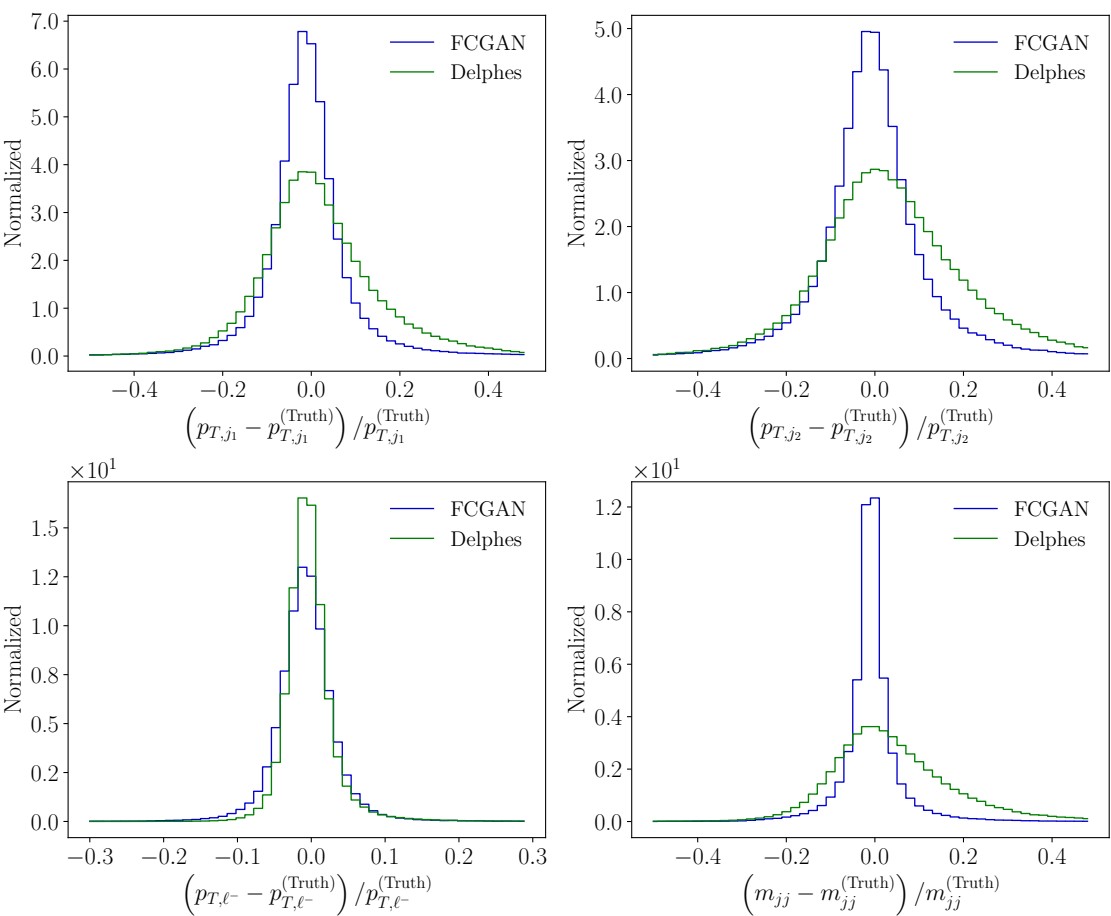

Figure 12: Normalized deviation between the FCGANned sample and truth (residual) for some of the kinematic variables.

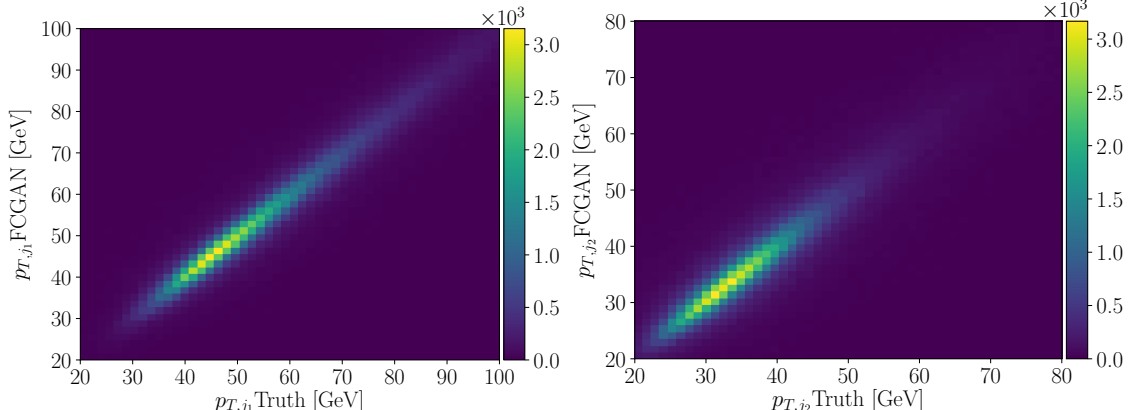

Figure 13: Correlations between the FCGAN-inverted and parton-level truth kinematics, or migration matrix.

truth kinematics and the DELPHES and FCGAN-inverted kinematics, for instance

$$\frac{p_{T,j}^{(\mathrm{FCGAN})} - p_{T,j}^{(\mathrm{Truth})}}{p_{T,j}^{(\mathrm{Truth})}} \qquad \text{and} \qquad \frac{p_{T,j}^{(\mathrm{DELPHES})} - p_{T,j}^{(\mathrm{Truth})}}{p_{T,j}^{(\mathrm{Truth})}}. \tag{15}$$

The events shown in these histograms correspond to the full phase space inversion shown in Fig. 6, but from the discussion in the main text it is clear that the picture does not change when we invert only part of phase space. As expected, we see narrow peaks around zero, with a width in the $\pm 10\%$ range for the jet momenta and much more narrow for the leptons, which are less affected by detector smearing. For all distributions, but especially the reconstructed $W$-mass, we see that the FCGAN reconstruction is significantly closer to the parton-level truth than the DELPHES events are.

Finally, we show the migration matrix or correlation between true parton-level and reconstructed parton-level events in terms of some of the kinematic variables in Fig. 13. Not surprisingly, we observe narrow diagonal lines.

# B   Staggered vs cooling MMD

The MMD loss is a two-sample test looking at the distance between samples $x$, $x'$, drawn independently and identically distributed, in terms of a kernel function $k(x, x')$. Implementations of such a kernel, as given in Eq. 6, include a fixed width or resolution $\sigma$. We employ the MMD loss to reproduce the invariant mass distribution of intermediate on-shell particles $M_p$. A natural choice of $\sigma$ is the corresponding particle width. However, this is inefficient at the beginning of the training, when any generated invariant mass $M_G$ is essentially a random uniform distribution. In that case $(x - x')^2 \gg \sigma^2$ for any $x, x' \sim M_G$, and Eq. 5 reduces to

$$\mathrm{MMD}(k; M_G, M_P) \simeq \sqrt{\langle k(y, y') \rangle_{y,y' \sim M_P}} \simeq \text{const}, \tag{16}$$

and provides little to no gradient.

This can be avoided by computing the MMD loss using multiple kernels with decreasing widths, so that the early training can be driven by wide kernels. A drawback of this approach is that only the small subset of kernels with a resolution close to the evolving width of $M_G$ gives a non-negligible gradient.

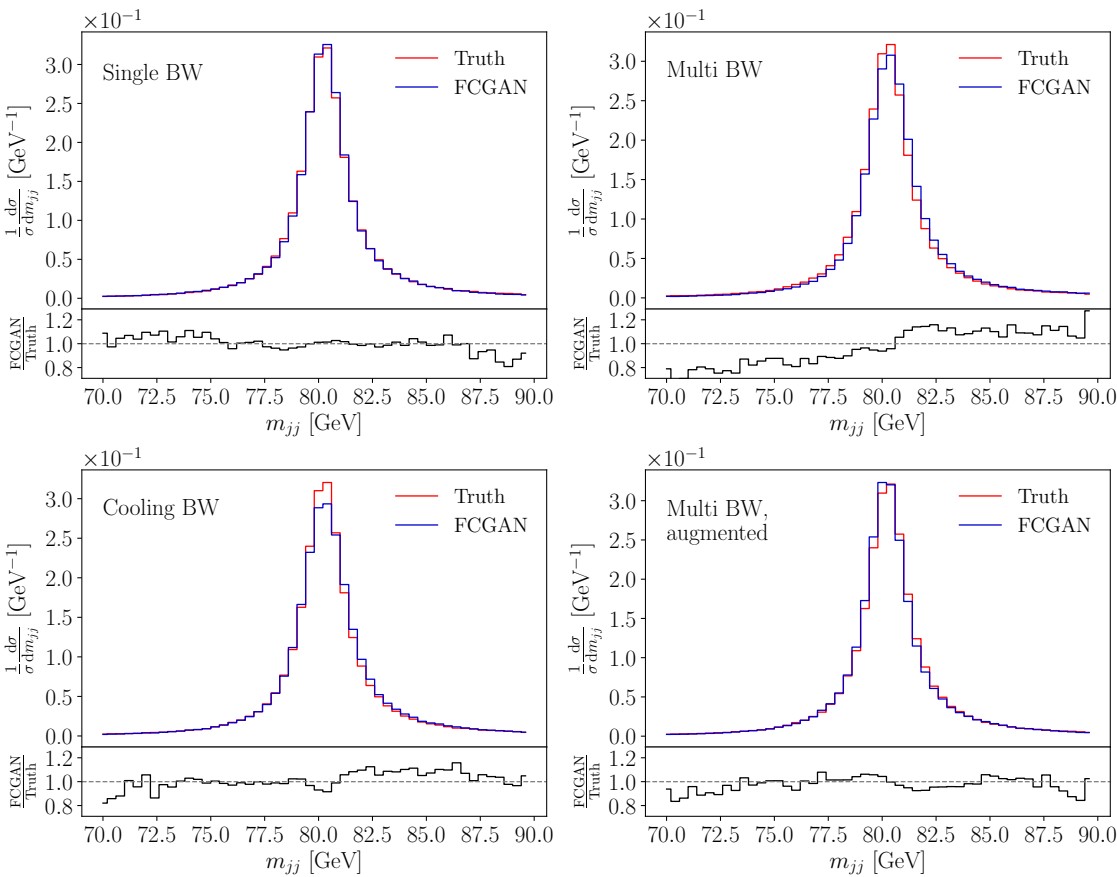

Figure 14: Invariant jet-jet mass distribution for different MMD loss implementations: single kernel (upper left), multiple kernels (upper right), cooling kernel (lower left) and augmented multiple kernels (lower right).

Alternatively, we can employ a cooling kernel, which we initialize to some large value and then shrink to the correct particle width. This is an efficient solution at all stages of the training. A subtlety is that the rate of the cooling has to follow the pace of the generator in producing narrower invariant mass distributions. Ultimately, we want to avoid hand-crafting the cooling process, because it adds hyper-parameters we need to tune. We use a dynamic kernel width as a fixed fraction of the standard deviation of the $M_G$ distribution. This standard deviation as an estimate of the width of $M_G$ can be replaced by any measure of the shape of $M_G$, such as the full width at half maximum, and our tests show that the performance is largely insensitive to the choice of the fraction.

Yet another approach is based on the observation that the MMD kernel test is not restricted to one-dimensional distributions [48,53,54]. This allows us to improve the invariant mass reconstruction by including additional physical information $x_i$ to the test, so that the discrepancy is not computed just between the samples $M_P$ and $M_G$ of real and generated invariant masses, but rather between $(M_P, x_P)$ and $(M_G, x_G)$. In the FCGAN spirit we therefore augment the batches of true and generated invariant masses with one of conditional invariant masses. From the same detector information used to condition the generator and the discriminator, we can extract the detector level invariant masses $M_D$ and accordingly compute $\text{MMD}(k; (M_G, M_D), (M_P, M_D))$. Even tough this does not represent a conditional MMD, training with multiple kernels benefits from using the augmented batches. In Fig. 14 we compare the same invariant mass distribution using these different MMD implementations.

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
