# Peer review of "How to GAN away Detector Effects"

_SciPost Physics, doi:SciPost Phys. 8, 070 (2020)_

## Round 2 · Referee Report · Anonymous · 2020-2-1

Report
This paper discusses the use of generative networks to deal with certain detector effects and argues that an event-level matching before and after detector simulation helps to reduce the bias in a number of observables, as illustrated for the case of semileptonic $WZ$ decays. The methodology is interesting and the presented case study serves as a nice proof of principle. However, the paper could really benefit from a revised introductory discussion, which fails to outline major assumptions and simplifications made by the authors. I will elaborate my concerns below and hope that a revised discussion can further improve the quality of the paper.
Requested changes
1. Unfolding methods are routinely used by experimenters to correct the data for detector effects in a way that aims to be minimally model dependent in order to allow for a direct comparison of the data to current and future theory calculations, without the need for computationally expensive detector simulation. To this end, the data are usually unfolded back to the *particle* level, constructed from physically meaningful colourless (i.e. fully hadronised) final-state particles. Correcting the data back to the quantum-mechanically ambiguous *parton* level would require a correction for non-perturbative effects in addition to the effects of the detector, thereby extrapolating over essentially everything we don’t fully understand about (soft) QCD. In order to ease the comparison of the particle-level data with more state-of-the-art fixed-order or resummed calculations, which are both at the parton level, the experiments might also publish additional non-perturbative correction factors, but they are (if at all) provided on a "use at your own risk" basis.
Presumably, this must be known to the authors and the decision to choose the extrapolation back to the parton level was just a pragmatic one. Nevertheless, I would advise that the associated caveats be acknowledged properly to avoid further criticism from unfolding experts in the community.
That said, the second-to-last paragraph of the introductory section seems to advocate an unfolding to the parton level as the preferred way to report measurement results. I can only hope the authors are not actually serious about this suggestion, which of course would severely reduce the usefulness of the data compared to current LHC cross-section measurements.
On the other hand, if the argument is meant to be directed at the LHC searches who typically report their results at the detector level, then I think that's a fair appeal to the experimental search groups, but also not entirely obvious from the current wording. Nevertheless, search results reported at the parton level would similarly be of limited use for reinterpretation purposes due to their intrinsic model dependence and so my criticism regarding parton vs particle level still applies.
2. Training a machine-learning algorithm to map between the detector-level and parton-level of a specific process is a technically interesting problem, but the setup presented in this paper is very simplified, brushing aside a lot of the difficulty encountered in a realistic unfolding problem.
The data are comprised of all sorts of processes contributing to a given final state and in a typical analysis most of the time is actually spent on understanding the reducible and irreducible background processes in order to correct the data for them. It is important to realise that these "background subtractions" can only be applied to the data on a stochastic basis. The data are effectively just scaled, but they remain a potpourri of different event topologies originating from different processes, all giving rise to the same final state of interest. The experimenter won’t have the luxury of a pure sample of $Z(\to\ell\ell)W(\to jj)$ events where a simple migration matrix — constructed from a single process — is sufficient to perform an unfolding. In reality, a lot of effort is also going into assessing the robustness of the chosen unfolding method against the unknown true process composition in the data and the analysis is optimised to reduce potential biases as much as possible.
Now, the methodology presented in this paper is mainly concerned with providing an alternative approach to correct for the bin-to-bin and out-of-acceptance migrations of the events in a given distribution, without really addressing the more laborious parts of the unfolding problem. There is much room for future studies in this direction of course, and so it would be good to at least acknowledge the simplifications made for this study to put things a little more into perspective.
3. If an experimental collaboration were to publish a paper saying they "generated events using Madgraph5", giving no additional information on the type of setup used whatsoever, they would be guaranteed to receive a deluge of abuse from their theory colleagues, well justified one might argue. In the interest of reproducibility at least, may I suggest you elaborate a little: What is the perturbative accuracy of your event sample? Is it a multi-leg setup? What parton shower was used? etc.
4. The event selection on page 4 is far too flimsy for my taste: Fiducial selection criteria are only specified for jets. What about the leptons? Presumably there would have to be an ATLAS-like lepton selection if you’re using the Delphes ATLAS card? Can you please state which charged lepton flavours you even consider? Can you also clarify whether the same fiducial selection is applied at both parton level and detector level?
5. On page 2, the acronym "GAN" is being introduced as "generative network", which then makes it a noun and so grammar aficionados will find the sentences where the acronym is being used as a verb somewhat painful to read ("we [generative network] only part of the phase space" for instance).
Author: Ramon Winterhalder on 2020-03-19 [id 770]
(in reply to Report 1 on 2020-02-01)
1) Unfolding methods are routinely used by experimenters to correct the data for detector effects in a way that aims to be minimally model dependent in order to allow for a direct comparison of the data to current and future theory calculations, without the need for computationally expensive detector simulation. To this end, the data are usually unfolded back to the particle level, constructed from physically meaningful colourless (i.e. fully hadronised) final-state particles. Correcting the data back to the quantum-mechanically ambiguous parton level would require a correction for non-perturbative effects in addition to the effects of the detector, thereby extrapolating over essentially everything we don’t fully understand about (soft) QCD. In order to ease the comparison of the particle-level data with more state-of-the-art fixed-order or resummed calculations, which are both at the parton level, the experiments might also publish additional non-perturbative correction factors, but they are (if at all) provided on a "use at your own risk" basis. Presumably, this must be known to the authors and the decision to choose the extrapolation back to the parton level was just a pragmatic one. Nevertheless, I would advise that the associated caveats be acknowledged properly to avoid further criticism from unfolding experts in the community. That said, the second-to-last paragraph of the introductory section seems to advocate an unfolding to the parton level as the preferred way to report measurement results. I can only hope the authors are not actually serious about this suggestion, which of course would severely reduce the usefulness of the data compared to current LHC cross-section measurements. On the other hand, if the argument is meant to be directed at the LHC searches who typically report their results at the detector level, then I think that's a fair appeal to the experimental search groups, but also not entirely obvious from the current wording. Nevertheless, search results reported at the parton level would similarly be of limited use for reinterpretation purposes due to their intrinsic model dependence and so my criticism regarding parton vs particle level still applies.
-> We are aware that most of ATLAS and CMS does not unfold to the parton level, and we have discussed with many experts. There are, however, some attempts to unfold top observable to the parton, or even top production level, and we have found them extremely useful for instance in a global EFT fit. So this is indeed the direction we are going, but we are of course aware that this direction needs work and convincing results. We added quite a bit of text to the introduction and to Sec.2 to make this clearer.
2) Training a machine-learning algorithm to map between the detector-level and parton-level of a specific process is a technically interesting problem, but the setup presented in this paper is very simplified, brushing aside a lot of the difficulty encountered in a realistic unfolding problem.
The data are comprised of all sorts of processes contributing to a given final state and in a typical analysis most of the time is actually spent on understanding the reducible and irreducible background processes in order to correct the data for them. It is important to realise that these "background subtractions" can only be applied to the data on a stochastic basis. The data are effectively just scaled, but they remain a potpourri of different event topologies originating from different processes, all giving rise to the same final state of interest. The experimenter won’t have the luxury of a pure sample of
Z(->ll)W(->jj)
events where a simple migration matrix — constructed from a single process — is sufficient to perform an unfolding. In reality, a lot of effort is also going into assessing the robustness of the chosen unfolding method against the unknown true process composition in the data and the analysis is optimised to reduce potential biases as much as possible.
Now, the methodology presented in this paper is mainly concerned with providing an alternative approach to correct for the bin-to-bin and out-of-acceptance migrations of the events in a given distribution, without really addressing the more laborious parts of the unfolding problem. There is much room for future studies in this direction of course, and so it would be good to at least acknowledge the simplifications made for this study to put things a little more into perspective.
-> Following this comment and discussions with others we have added a new section, where we inject new physics events with an s-channel resonance topology into the test sample. We hope that our results count as a first and promising step in the direction of model independence. We thought about changing the Monte Carlo generator, but given that we truncate our number of jets and start with jet-level observables that would not have been a useful test.
3) If an experimental collaboration were to publish a paper saying they "generated events using Madgraph5", giving no additional information on the type of setup used whatsoever, they would be guaranteed to receive a deluge of abuse from their theory colleagues, well justified one might argue. In the interest of reproducibility at least, may I suggest you elaborate a little: What is the perturbative accuracy of your event sample? Is it a multi-leg setup? What parton shower was used? etc.
-> We now give more details about how the events are generated: at which order, which parton shower, which jet algorithm etc.
4) The event selection on page 4 is far too flimsy for my taste: Fiducial selection criteria are only specified for jets. What about the leptons? Presumably there would have to be an ATLAS-like lepton selection if you’re using the Delphes ATLAS card? Can you please state which charged lepton flavours you even consider? Can you also clarify whether the same fiducial selection is applied at both parton level and detector level?
-> We now specify that we are using electrons, and we are applying the fiducial cuts at the detector level. The selection is meant to start flimsy, since we are beefing it up quite a bit later.
5) On page 2, the acronym "GAN" is being introduced as "generative network", which then makes it a noun and so grammar aficionados will find the sentences where the acronym is being used as a verb somewhat painful to read ("we [generative network] only part of the phase space" for instance).
-> We are aware of this, but we trust the authors to understand this definition. We devoted a detailed footnote to the topic in an earlier paper, but did not want to just copy it over.
Andy Buckley on 2020-01-18 [id 713]
I read this with interest, although did not have the time to do so in full detail and hence this is an informal comment rather than official review report. It's technically rather nice, although I was under the impression that unfolding has been attempted and achieved with ML methods several times before, and it would be good to highlight the distinctions of this version.
It would also be good to engage with the physics and the fundamental statistics of event calculation and detector interaction. The claim to unfold to parton level will ring alarm bells for many who note that the degrees of freedom at parton level in an event generator record are usually not 100% physical, and so any such extraction will unavoidably bear biases from calculation schemes, as well as any semi-arbitrary choices in how a given generator represents its intermediate stages. A more careful choice of unfolding target would be a safely reconstructed event topology from final-state objects, avoiding the physical-ambiguity implications of a parton-level target. The hadronization process is usually recognised as the level that unfolding should not attempt to extrapolate beyond, the decay and detector interaction stages following it being regarded as sufficiently classical to be safe (as opposed to the QM dynamics of the partonic bit that precedes it). Maybe this is all known and the parton level used for convenience despite its issues -- which seems a trivial reason to introduce a whole avenue of criticism --but if so then please make the straw-man nature of the target clear in the text.
Of course, using final-state MC observables still introduces model dependence via the generator (which, by the way, isn't clearly specified beyond "using MadGraph5" -- with merging? how many extra partons? what parton shower and tune?) and the detector model. In practice, neither is a perfect match to data to be unfolded, and most time spent on unfolding goes into assessing non-closures due to model variations, rather than the estimation of a migration matrix that is the equivalent of the method discussed here. It seems like GANs could potentially help with parametrising systematics, too, but these issues at least need to be acknowledge if not actually addressed.
Finally, it seems that the GAN approach generates distributions that "look right", which is fine probabilistically... indeed to my mind it would be better to explicitly generate hundreds or thousands of GAN events per reco event, to map its likelihood distribution. But a single mapping of reco to truth event through the GAN is not "the answer" -- in general such a thing cannot be known by any method, even without the "parton level" issue. Maybe it represents the maximum-likelihood truth configuration for that reco event, or is just a single sample from the conditional likelihood distribution? Since there is room for confusion here, and potentially for calamity if a one-for-one GANning of reco events were attempted in a low-stats search region, it would be good also to discuss this "philosophical" issue of what unfolding means in the context of your method.

---

## Round 2 · Referee Report · Anonymous · 2020-2-5

Report
The article "How to GAN away Detector Effects" describes a method to unfold detector effects on measured differential distributions using generative networks trained on Monte Carlo simulation.
In particular they show how using event-by-event information at parton-level and after detector simulation a bias due to the acceptance can be avoided.
This is, to my knowledge, a novel technique which certainly deserves to be published in this journal.
Below you can find some questions/comments on a few points that I found more difficult to read and understand, and some mistypings. As a general commeny I can add that my main difficulty is understanding the meaning and usage of MMD in the context of this work, but I will come to that point while going through the text.
page 2, just before (1): in “(1.FCGAN)” remove “1.”
page 2, last line: “external masses”: what is the meaning of “external” here?
page 3: Just before eq. (5) you write, about MMD you write “It allows us to compare pre-defined distributions, for instance the one-dimensional invariant mass of an intermediate particle”. So, the variables x and y in eq. (5) are masses or they can be anything? And if they can be anything, which ones did you choose? I think an example here would be useful. Maybe I am confused by the choice to use Gaussian or Breit-Wigner kernel functions, which made me think of a resonance. But the MMD is a general statistical test and therefore can be applied to anything, if I understood it correctly. Some further clarifications here would be very useful.
page 4: why \lambda_D << \lambda_G? Is there a clear motivation?
page 4: which MC parton shower did you use with Madgraph5?
page 4: “statistically independent, but otherwise identical sets of detector-level events” means an independent sample drawn from the same population of the one used for the training? If that is the case, I think the sentence is not very clear. I could understand it only after I reached the point where the unfolding is applied to a sample with different acceptance cuts.
page 4, last paragraph: concerning the batch size, were you limited to 512 by the memory size? Some additional information on the hardware setup you used would be an added value to the paper.
What do you mean by “the matching requirement”? You just wrote that batches are “independently chosen”, so I understood that there is no matching requirement.
page 5: the cuts in (7) and (8) are applied at detector level: in figure 4, the corresponding original parton-level distribution correspond to this acceptance at detector level. Even if it is interesting to check if GAN-inverted distributions reproduce the parton-level ones, from a practical point of vue it would be more useful to unfold measurements to a well defined phase space at parton-level, i.e. a fiducial region for the measurement. Maybe it is a different problem and requires a different approach, but have you considered it?
page 6: “a classification network could be improved through a variational feature in latent space”: can you add a reference? “a standard solution”: why “standard”?
page 7, eq. (9): “P_T” has not been defined. Should it be “P_p”?
page 7: “we do not build a conditional MMD loss”: does it mean that MMD is unchanged with respect to section 2 GAN or that it is not used (and then should be removed from figure 5)?
page 8: what does it mean “at the 90% level”? do you mean that differences are < 10%?
page 10: “the MMD loss is is not actually conditional”: remove an “is”.
page 10: “the standard implementations are somewhat inefficient”: can you give a reference for the “standard implementations”?
page 14: “implementations of such a the kernel”: remove “the”
page 15, first line: “A subtlety is is that”: remove an “is”
page 15: can you clarify better, with an example, what does it mean: “we augment the batches of true and generated invariant masses with one of conditional invariant masses, computed from the same detector information used to condition the generator and the discriminator”?
page 17: in ref 22 there is a mis-typing.
Best regards
Author: Ramon Winterhalder on 2020-03-19 [id 769]
(in reply to Report 2 on 2020-02-05)
1) page 2, just before (1): in “(1.FCGAN)” remove “1.”
-> Done.
2) page 2, last line: “external masses”: what is the meaning of “external” here?
-> Now specified.
3) page 3: Just before eq. (5) you write, about MMD you write “It allows us to compare pre-defined distributions, for instance the one-dimensional invariant mass of an intermediate particle”. So, the variables x and y in eq. (5) are masses or they can be anything? And if they can be anything, which ones did you choose? I think an example here would be useful. Maybe I am confused by the choice to use Gaussian or Breit-Wigner kernel functions, which made me think of a resonance. But the MMD is a general statistical test and therefore can be applied to anything, if I understood it correctly. Some further clarifications here would be very useful.
-> A detailed discussion about the MMD in our GAN setup is available as 1907.03764, we now mention this in a footnote.
4) page 4: why \lambda_D << \lambda_G? Is there a clear motivation?
-> These are just tunable hyper-parameters, and the values just chosen by optimization.
5) page 4: which MC parton shower did you use with Madgraph5?
-> We used Pythia8, with the standard Pythia8 card but with inital-state radiation turned off, as now specified in the text.
6) page 4: “statistically independent, but otherwise identical sets of detector-level events” means an independent sample drawn from the same population of the one used for the training? If that is the case, I think the sentence is not very clear. I could understand it only after I reached the point where the unfolding is applied to a sample with different acceptance cuts.
-> We now define them as `statistically independent, but simulation-wise identical', to make it more clear.
7) What do you mean by “the matching requirement”? You just wrote that batches are “independently chosen”, so I understood that there is no matching requirement.
-> Now clarified in the text.
8) page 5: the cuts in (7) and (8) are applied at detector level: in figure 4, the corresponding original parton-level distribution correspond to this acceptance at detector level. Even if it is interesting to check if GAN-inverted distributions reproduce the parton-level ones, from a practical point of vue it would be more useful to unfold measurements to a well defined phase space at parton-level, i.e. a fiducial region for the measurement. Maybe it is a different problem and requires a different approach, but have you considered it?
-> We have not considered parton-level cuts at this stage, but we agree they would be an interesting question especially when they lead to sharp structures. However, we prefer for open that rather technical topic in a possible follow-up.
9) page 6: “a classification network could be improved through a variational feature in latent space”: can you add a reference? “a standard solution”: why “standard”?
-> We now explicitly mention the upgrade for instance from autoencoders to variational autoencoders as a standard concept.
10) page 7, eq. (9): “P_T” has not been defined. Should it be “P_p”?
-> Thank you for point this out, it was a remainder of 'truth' rather than 'parton level'.
11) page 7: “we do not build a conditional MMD loss”: does it mean that MMD is unchanged with respect to section 2 GAN or that it is not used (and then should be removed from figure 5)?
-> Changed to `we do not build a conditional version of the MMD loss'
12) page 8: what does it mean “at the 90% level”? do you mean that differences are < 10%?
-> Yes, now clarified.
13) page 10: “the MMD loss is is not actually conditional”: remove an “is”.
-> Done
14) page 10: “the standard implementations are somewhat inefficient”: can you give a reference for the “standard implementations”?
-> We removed the `the' to indicate that there is really no good standard implementation worth mentioning or citing...
15) page 14: “implementations of such a the kernel”: remove “the”
-> Done
16) page 15, first line: “A subtlety is is that”: remove an “is”
-> Done
17) page 15: can you clarify better, with an example, what does it mean: “we augment the batches of true and generated invariant masses with one of conditional invariant masses, computed from the same detector information used to condition the generator and the discriminator”?
-> We updated and expanded the description of this approach and hope it is clear now.
18) page 17: in ref 22 there is a mis-typing.
-> We are sorry, but we did not find this typo, please let us know and we will remove it.

---

## Round 2 · Referee Report · Anonymous · 2020-2-7

Report
The manuscript "How to GAN away Detector Effects" presents a novel method to unfold detector effects without significantly biasing the corrected distributions. It is highly valuable, and generally well written.
I have a few comments I would like to be addressed though.
Requested changes
1) The authors write they generate the events with MadGraph5 and detector simulate it with Delphes. It should also be stated which tools they use to simulate the parton shower evolution, multiple interactions, hadronisation, hadron decay and QED corrections that lie in between these two steps in a full event simulation chain, and which they surely include in their simulation, as detector responses to partons are not a sensible concept.
2) The authors generally refer to distributions at parton level and detector level. Unless, they are really referring to particle level (ie. particles that are stable and propagate freely on scales of c*tau about 1cm), the proposed unfolding procedure conflates two physically very separate effects: the above transition from short-distance partons into long-distance somewhat stable leptons, photons and hadrons, and the interaction of the latter with the detector material. Please comment and rename if appropriate.
3) In Section 2, how dependent are the results on the precise setup of batch size, the number of epochs and iterations per epochs. The authors did probably check this in detail, to my experience, the number of epochs vs the number of points per epoch seems ill-balanced. I would have expected significantly less epochs with significantly more points per epoch to not be affected by statistical effects for each epoch.
4) The authors focus on mostly on hadronic observables which should receive the largest detector corrections. Besides the lepton transverse momentum, which the authors show, it would be interesting to know how well the dilepton invariant mass is reproduced by the proposed unfolding techniques. It would give a glimpse into how well the method could perform for precision observables.
5) The authors correctly note that their method, when applied to only a moderately small subset (a fraction of 40%) can fail for selected distributions. While I want to strongly commend the authors to show the limitations of their method, I would like to ask them to comment on whether it can be anticipated under which circumstances the unfolding can or will fail. This is especially important as once it is applied to real data, the truth is of course unknown.
6) The fact that it unfolds the dijet-invariant mass distribution so differently under cut III and cut IV, does that imply that good unfolding of the same distribution under the more inclusive cuts I and III is largely accidental? In the sense that subsets that enter the distribution in the more inclusive case are unfolded so much worse when applied to only the subset.
7) I would like to encourage the authors to not use nouns as verbs, eg. to GAN something is ill-defined. It is akin to stating "pocket-calculator an equation". Please, in each such case, use verbs to describes what one is supposed to do with the GAN (or else) to the respective object.
Author: Ramon Winterhalder on 2020-03-19 [id 768]
(in reply to Report 3 on 2020-02-07)
1) The authors write they generate the events with MadGraph5 and detector simulate it with Delphes. It should also be stated which tools they use to simulate the parton shower evolution, multiple interactions, hadronisation, hadron decay and QED corrections that lie in between these two steps in a full event simulation chain, and which they surely include in their simulation, as detector responses to partons are not a sensible concept.
-> We now mention that we use the standard Pythia8 shower etc, and cite the tool chain accordingly.
2) The authors generally refer to distributions at parton level and detector level. Unless, they are really referring to particle level (ie. particles that are stable and propagate freely on scales of c*tau about 1cm), the proposed unfolding procedure conflates two physically very separate effects: the above transition from short-distance partons into long-distance somewhat stable leptons, photons and hadrons, and the interaction of the latter with the detector material. Please comment and rename if appropriate.
-> As discussed above, we now clarify in the Introduction and in Sec.2 that we indeed mean parton level (as defined by the hard process in the MC simulation) and DELPHES.
3) In Section 2, how dependent are the results on the precise setup of batch size, the number of epochs and iterations per epochs. The authors did probably check this in detail, to my experience, the number of epochs vs the number of points per epoch seems ill-balanced. I would have expected significantly less epochs with significantly more points per epoch to not be affected by statistical effects for each epoch.
-> A large batch size is of course computationally expensive, but is very much required to have a good mapping of the invariant mass peak, the one we chose is a good compromise between these two. With respect to the number of epochs, a rough estimate is that with a batch size of 512, 300 points per epoch means covering half the training set every epoch, which is perfectly fine with such a slow learning rate decay. Finally, we agree that the number of epochs is absolutely oversized and unneccesary, (as is the number of parameters of the model). However, it is clear that this paper is not focused on optimizing GANs.
4) The authors focus on mostly on hadronic observables which should receive the largest detector corrections. Besides the lepton transverse momentum, which the authors show, it would be interesting to know how well the dilepton invariant mass is reproduced by the proposed unfolding techniques. It would give a glimpse into how well the method could perform for precision observables.
-> We agree that in principle this is corrct, but our network trivially reproduces the unity transformation at least at the level we are showing right now.
5) The authors correctly note that their method, when applied to only a moderately small subset (a fraction of 40%) can fail for selected distributions. While I want to strongly commend the authors to show the limitations of their method, I would like to ask them to comment on whether it can be anticipated under which circumstances the unfolding can or will fail. This is especially important as once it is applied to real data, the truth is of course unknown.
-> We added a sentence stating that the network first fails in mjj, because it is not very stable, while all the other variables are still okay.
6) The fact that it unfolds the dijet-invariant mass distribution so differently under cut III and cut IV, does that imply that good unfolding of the same distribution under the more inclusive cuts I and III is largely accidental? In the sense that subsets that enter the distribution in the more inclusive case are unfolded so much worse when applied to only the subset.
-> We hope that the new Section 4 with the injected new physics signal shows that it is not accidental.
7) I would like to encourage the authors to not use nouns as verbs, eg. to GAN something is ill-defined. It is akin to stating "pocket-calculator an equation". Please, in each such case, use verbs to describes what one is supposed to do with the GAN (or else) to the respective object.
-> Yeah, but we think it's funny and adds to our street creds, so this is the one point where we would rather not follow the referee.

---

## Round 4 · Referee Report · Anonymous (Referee 1) · 2020-3-29

Report
I'd like to thank the authors for their responses and the clarifications in the revised version. I appreciate the extra test of the robustness of the method in the presence of signatures that are significantly different from the Standard Model. I think that makes for a nice addition to the paper.

---

## Round 4 · Referee Report · Anonymous (Referee 2) · 2020-4-11

Report
I am satisfied with authors replies to my previous comments.
The paper overall has improved (also thanks to other referees comments) and I think it is already in good shape to be published.
I appreciated in particular the new section 4 "New physics injection": it is not mandatory, but if there will be a new submission, I think it would be nice to see also the detector distributions in Figure 10 (in particular I am interested to see it for m_lljj, where I wonder if the unfolded new resonance, without additional constraints, improves its significance over the detector level one).
The hyperlink is missing in Ref 23: G. Cowan, Conf. Proc. C0203181 (2002) 248. [,248(2002)]
(also "[,248(2002)]" should not appear)
Many congratulations to the authors for this very nice and inspiring work.
The paper overall has improved (also thanks to other referees comments) and I think it is already in good shape to be published.
I appreciated in particular the new section 4 "New physics injection": it is not mandatory, but if there will be a new submission, I think it would be nice to see also the detector distributions in Figure 10 (in particular I am interested to see it for m_lljj, where I wonder if the unfolded new resonance, without additional constraints, improves its significance over the detector level one).
The hyperlink is missing in Ref 23: G. Cowan, Conf. Proc. C0203181 (2002) 248. [,248(2002)]
(also "[,248(2002)]" should not appear)
Many congratulations to the authors for this very nice and inspiring work.

---

## Editorial Decision

published